# A highly attenuating and frequency tailorable annular hole phononic crystal for surface acoustic waves

B.J. Ash[1], S.R. Worsfold[1], P. Vukusic[1] & G.R. Nash [1]

Surface acoustic wave (SAW) devices are widely used for signal processing, sensing and increasingly for lab-on-a-chip applications. Phononic crystals can control the propagation of SAW, analogous to photonic crystals, enabling components such as waveguides and cavities. Here we present an approach for the realisation of robust, tailorable SAW phononic crystals, based on annular holes patterned in a SAW substrate. Using simulations and experiments, we show that this geometry supports local resonances which create highly attenuating phononic bandgaps at frequencies with negligible coupling of SAWs into other modes, even for relatively shallow features. The enormous bandgap attenuation is up to an order-of-magnitude larger than that achieved with a pillar phononic crystal of the same size, enabling effective phononic crystals to be made up of smaller numbers of elements. This work transforms the ability to exploit phononic crystals for developing novel SAW device concepts, mirroring contemporary progress in photonic crystals.

---

[1] College of Engineering, Mathematics and Physical Sciences, University of Exeter, Exeter EX4 4QF, UK. Correspondence and requests for materials should be addressed to G.R.N. (email: G.R.Nash@Exeter.ac.uk)

The field of phononic crystals (PnCs) has attracted considerable attention in recent years as a method to control and manipulate acoustic and elastic wave propagation. PnCs are periodic elastic composites that can exhibit phononic bandgaps[1–3], a desirable dispersion property which can be used to design functional materials, for example, as acoustic transmission filters. The experimental confirmation of PnC bandgaps and their use for acoustic confinement was first found for bulk acoustic wave (BAW) systems with solid/solid and solid/fluid composites as the PnC, where it was found that the functionality of the structures were dependent on the optimisation of the filling fraction as well as the elastic constants of the constituent materials[4–8]. PnCs can also be designed for surface acoustic wave (SAW) systems, where the wave is an elastic displacement confined to within a wavelength of the surface of a solid, using the same principles, a characteristic particularly important for technologically-relevant applications[9, 10]. SAW devices, for example, are used as RF signal processors which are commonly found in the telecommunications industry[11, 12], where the use of frequency-dependant confinement could lead to increased functionality of devices mediated, for example, by acoustic cavities or waveguides. PnCs have also been shown to be a potential route toward programmable micro-fluidics[13].

A commonly used design for solid/solid PnC systems comprises a periodic array of etched holes that create bandgaps for propagating acoustic waves mediated by destructive Bragg conditions. These have been widely used for Lamb wave studies for which holes can be etched through the total depth of thin plates[14]. Holes of finite depth in thick substrates have also been used for SAW[15, 16] PnCs, however, the bandgaps achieved by these systems are at frequencies where leaky SAWs are dominant. This leads to a quick loss of amplitude with propagation distance as leaky SAWs are comprised of evanescent modes that couple to BAWs in the substrate. Coupling to BAWs occurs for SAWs outside of the soundline, where the soundline is analogous to the lightline in photonic systems, and is the linear dispersion of BAWs in an homogenous bulk medium. Low frequency bandgaps/energies, well below the soundline, are therefore desirable for SAW PnCs and have been demonstrated experimentally using locally resonant structures, such as pillars, to provide an additional mechanism to open bandgaps at the resonant frequencies[17–22]. Periodic array pillar PnCs have been used to demonstrate superlensing through negative refractive indices[23] and acoustic confinement through waveguiding[24]. Furthermore, random arrays of pillars have been confirmed to exhibit bandgap-like extinction of propagating SAWs[25].

However, PnCs comprised of pillar arrays suffer from a number of disadvantages. In particular, as the pillars obtrude from a surface they are structurally fragile and also, in contrast to non-resonant holes, or the recently demonstrated monolithic phonon-polariton induced PnCs[26], they are not compatible with the integration of other layers on the surface. For example, there is growing interest in the integration of SAW devices with 2D materials such as graphene and molybdenum disulphide[27, 28].

In this work we introduce a SAW PnC geometry consisting of annular holes of finite depth. These annular holes support locally resonant bandgaps, analogous to pillar-based geometries, but improve upon them in a potentially transformative way. This architecture provides both structural integrity and the potential for future integration of other layers. In addition, the bandgap attenuation per unit feature is up to an order-of-magnitude greater than for pillars, allowing effective PnCs to be realised with a much lower number of elements. The annular hole system also has an additional degree of geometric freedom compared to a

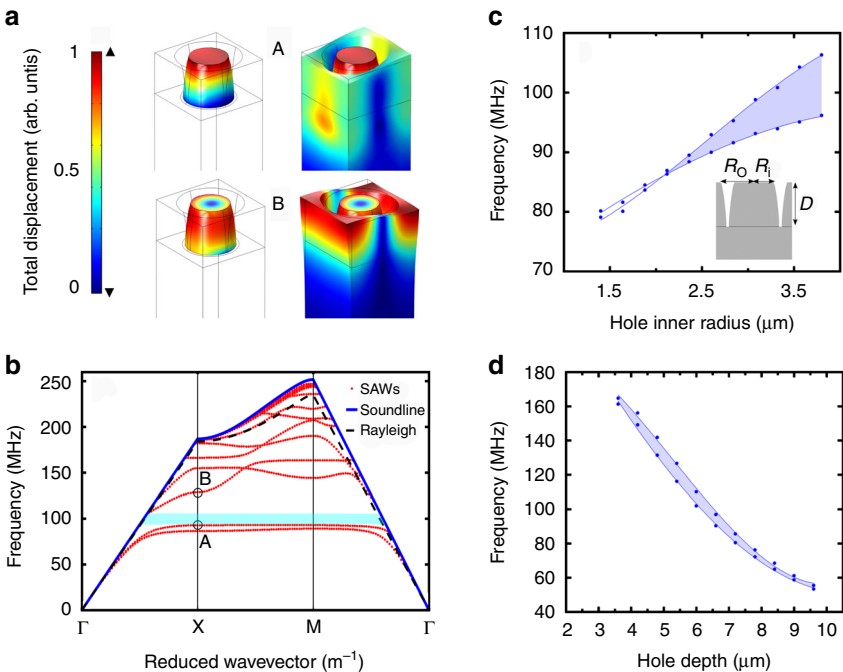

**Fig. 1** Simulated PnC band diagram, displacement profiles and tailorability plots. **a** Total displacement surface plots for modes A and B for the entire unit cell and for only the locally resonant structure. **b** SAW band diagram generated through finite element method simulation of annular hole unit cell with pitch 10.9 μm, inner radius 3.1 μm, outer radius 5.1 μm and hole depth 6.4 μm. The *blue line* represents the soundline and the *red lines* represent propagating SAW bands within the 1st Brillouin zone. The *teal bar* represents the PnC complete bandgap. **c** Bandgap limits of annular hole PnC against inner radius of the holes with cross section schematic of unit cell, with $R_O$, $R_i$ and $D$ being the outer radius, inner radius and depth of the annular hole respectively. The data points represent upper and lower frequency limits of the complete bandgap shaded in *blue*. The lines are polynomial fits to guide. **d** Bandgap limits against depth of the holes

cylindrical pillar as it has two tailorable radii instead of one. This allows for enhanced tailoring of the PnC SAW dispersions for both frequency and velocity. Finally, although annular holes have previously been considered for prospective photonic crystals, they have not been extensively exploited because the associated fabrication is not as straightforward as it is for acoustics[29, 30]. In our annular hole PnCs, the resonating structure also plays an active role in the SAW dispersion control, which it does not in the photonic case.

## Results

**Finite element method simulations.** Phononic dispersions of the annular hole PnCs were calculated using finite element method simulations in order to confirm the validity of the design and to investigate the band displacement profiles. This allowed design of PnCs that have bandgaps not only at frequencies that can be experimentally characterised, but also at frequencies in which the key feature sizes can be realised through standard fabrication techniques such as focussed ion beam (FIB) etching.

In the three-dimensional (3D) unit cell model the annular holes are given a conical depth profile to approximate achieved fabrication results and sit on an extended homogenous domain assumed to be a semi-infinite substrate. All domains are assumed to be the same 128° YX-LiNbO$_3$ cut that was used experimentally; this is a commonly used piezoelectric where the rotated crystal orientation maximises efficiency of SAW excitation through interdigital transducers. Using an eigenvalue frequency domain solver SAW solutions were found, such as those in Fig. 1a, showing the displacement profiles as a colour scale within the unit cell of two such solutions.

Phononic dispersions were found for swept wavevectors within the first Brillouin zone of the PnC square array in Fig. 1b. Here, an annular hole depth of 6.4 μm, inner radius of 3.1 μm, outer radius 5.1 μm and pitch of 10.9 μm is simulated. The centre of the Brillouin zone and boundaries in the [100] and [110] directions of the square array are annotated as the Γ, X and M points respectively. The blue and red lines represent the soundline and SAW bands respectively. Leaky SAW solutions are not shown. It is clear from this plot that a complete bandgap shaded in teal has been created at a low frequency with observable propagating SAW modes for frequency ranges both above and below the gap within the soundline. This feature has not been previously found for finite depth SAW PnCs comprising holes. As expected, the reason for the creation of these low frequency gaps is due to local resonances within the PnC unit cell, reducing the phononic group velocities through the PnC for specific bands in a similar manner to cylindrical pillar PnCs[23].

Surface plots of total displacement for two SAW modes that support local resonances, for all unit cell domains and for isolated resonant domains, are plotted in Fig. 1a. The position of these modes are annotated as A and B on the band diagram showing that they correspond to the bands that limit the complete bandgap illustrated. For the lower bandgap limit it can be seen that the local resonance has a clear depth dependence and for the upper bandgap limit a radial dependence, suggesting that the bandgap limits are tailorable through the two parameters. Figure 1c, d demonstrate this tailorability, showing the frequency range of the lowest bandgap for varying depths and inner radii of the annular holes. With increasing radius the width of the bandgap increases and with increasing depth the centre frequency of the bandgap decreases. Supplementary Fig. 1 shows the equivalent dispersion and tailorability for an analogous pillar PnC.

To quantify the attenuation of the annular holes, the transmission of SAWs through the PnC at swept frequencies was simulated for a finite number of structures. The transmission

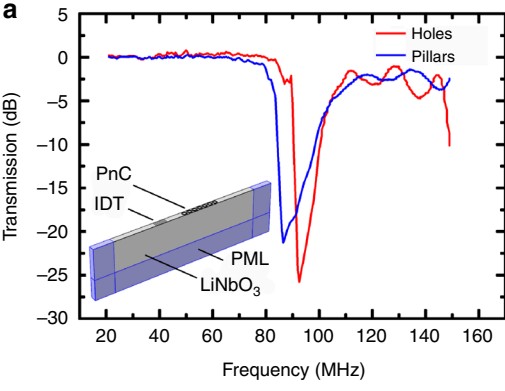

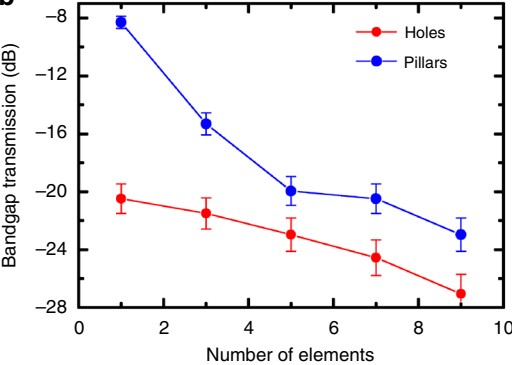

**Fig. 2** Simulated SAW transmission for novel annular holes and typical pillars. **a** Transmission vs. frequency for SAWs showing a deeper bandgap dip in transmission through 7 annular holes compared to 7 pillars. Inset schematic shows the PnC, interdigital transducer (IDT), LiNbO$_3$ substrate and shaded perfectly matched layers (PML) of the simulation. **b** Bandgap transmission minimum for increasing number of elements in propagation direction with s.d. error bars

through the PnC was found by measuring the average acoustic intensity over a defined area after SAW propagation and calculating the relative transmission compared to simulations of a blank substrate. Figure 2a shows this transmission for an annular hole PnC with 7 structures and a typical pillar PnC with the same unit cell dimensions as the holes. A value of close to zero here represents transmission similar to that of a blank substrate, and this is present over the frequency range for holes apart from at two distinct minima located between 85–110 MHz and above 145 MHz. Referring to the band diagram in Fig. 1b, at these frequencies there is either a bandgap or the PnC dispersion is diverging away from the Rayleigh line, so this reduction in transmission is expected and validates the model. Comparing the transmission between the holes system and pillars system, the same features are present in both spectra, however, the bandgap for the pillar system is offset in frequency by −10 MHz and the minimum transmission for the pillars is −19.0 dB compared to the holes much lower −24.5 dB. Figure 2b shows the minimum bandgap transmission vs. the number of elements and demonstrates that annular holes consistently outperform pillars in relation to bandgap attenuation efficiency across the range. In particular, the annular holes are far more effective at inducing bandgap attenuation with very few numbers of elements, where on a linear intensity scale, in the extreme case, transmission through 1 annular hole is 16 times lower than transmission through 1 pillar.

We propose that the reason for this dramatically improved bandgap extinction is because the local resonances within the

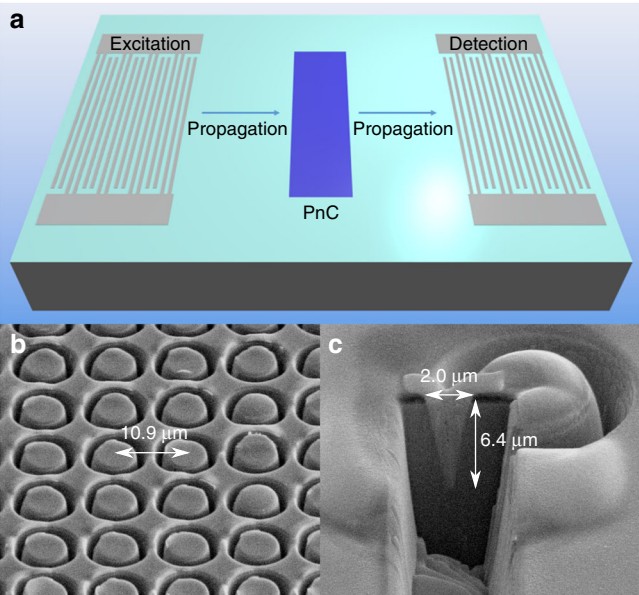

**Fig. 3** SAW delay line and fabricated PnC. **a** Schematic of device used to characterize the PnC with excitation and detection inter-digital transducers and propagation direction. The *blue* shaded area represents the fabricated PnC which covers the transducer aperture. **b** Section of a 3 mm × 80 μm square array of annular holes with depth 6.4 μm, inner radius 3.1 μm, outer radius of 5.1 μm and pitch 10.9 μm. **c** Cross section depth profile of an individual annular hole with platinum within the hole to provide image contrast

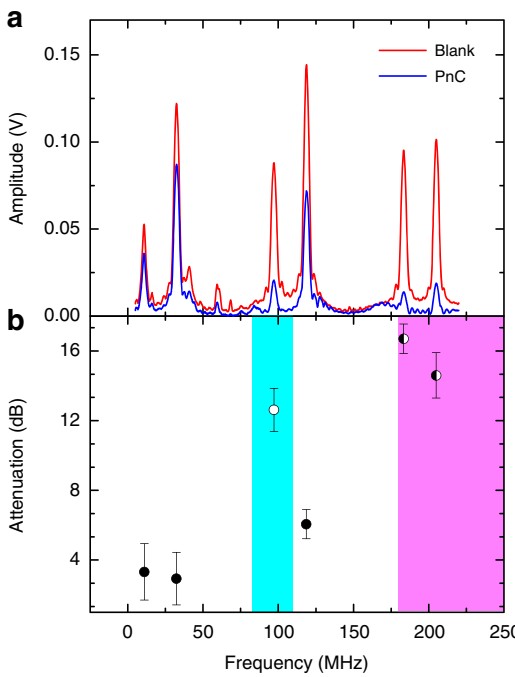

**Fig. 4** Experimental SAW transmission through PnC. **a** Transmission spectra of SAWs excited and detected through inter-digital transducers (IDTs) for a device with an unpatterned, blank substrate and with a 3 mm × 80 μm square array of finite depth annular holes as a PnC. **b** SAW attenuation of propagating SAWs at each resonant IDT peak relative between a blank substrate and the fabricated PnC, calculated as $20 \log(A_{\text{patterned}}/A_{\text{blank}})$, with s.d. error bars. Solid on *white*, empty on *blue* and half-filled on *pink* markers represent frequencies of expected propagating SAW bands, bandgap and leaky SAWs respectively

annular hole structure are more strongly excited than for an associated pillar structure. This arises from the fact that the incident SAW energy is confined to within 1 wavelength of the surface, meaning that the interface area extends to the depth of the holes and is therefore much larger than the SAW-pillar interface. This is evidenced in the transmission simulations where the average displacement within the annular holes is two orders of magnitude larger than that within the pillars.

These results show that 7 pillars are required to match the equivalent attenuation of 1 annular hole, a feature that demonstrates a significant advantage for PnC applications where a bandgap is required, such as for acoustic cavities or waveguides. This means that a smaller area or number of periods would need to be fabricated, thereby reducing device sizes and/or improving device performance.

In the above results the unit cell geometry of 6.4 μm height, 4.1 μm centre radius was considered as this structure has a height to depth aspect ratio of 1.56 which can be easily fabricated and is comparable to typical pillar geometries previously studied[18, 19, 21, 25]. Supplementary Table 1 shows a comparison of the performance of our annular holes and pillars for 3 differing aspect ratios, demonstrating that the annular hole PnC leads to a much higher bandgap attenuation than the pillar PnC in all but the most extreme, and impracticable, geometry. As the relationship between bandgap attenuation and aspect ratio is found to be complex and non-linear, we propose a thorough investigation for future studies.

**Device fabrication.** To validate the novel geometry experimentally, annular holes were patterned on the surface of commercially available 128° YX-LiNbO₃ SAW delay line devices with pre-fabricated transducers. To fabricate holes of predictable depth profiles in hard anisotropic materials such as LiNbO₃ a dry etching method is needed[31, 32]. A major advantage of our annular hole design is that it requires a significantly small amount of

substrate volume to be etched compared to pillars or even conventional holes. As a result of this, fabrication methods that are not feasible for other structures can be used, such as the FIB etching approach used in this work. The sample preparation needed is limited to deposition of a conducting layer that was grounded during etching and removed afterwards. A square array of annular holes of depth 6.4 μm, inner radius 3.1 μm, outer radius of 5.1 μm and pitch 10.9 μm, as simulated above, was patterned over an area of 3 mm × 80 μm in order to cover the entire transducer aperture, as shown schematically in Fig. 3a. Figure 3b, c shows oblique scanning electron microscope images of a magnified portion of the annular hole array and cross-sectioned depth profile, demonstrating that all the holes are uniform, equally spaced and are well approximated by the unit cell geometry used in simulations. The non-flat surface in the two images is due to the resist layer separating the conducting layer from the LiNbO₃ substrate being exposed to charge and 'bubbling'; however, before characterisation these two layers are removed from the substrate.

**Electrical measurements.** In Fig. 4a the measured SAW amplitude is plotted as a function of frequency for both annular hole patterned devices and blank devices under vacuum (a comparison to a pillar device is not possible here because the fabrication method cannot be used to produce such a PnC). Two identical, uniform double-electrode input/output transducers are used for SAW excitation and detection, as shown in Fig. 3a, which are designed to give resonances at the discrete frequencies shown. Here the resonances in measured amplitude at 11 and 33 MHz correspond to the fundamental modes of the transducers and the

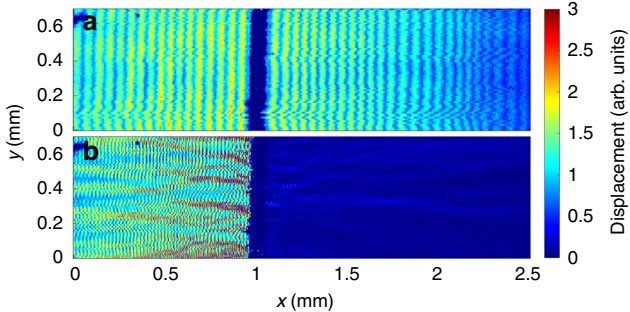

**Fig. 5** Measured out of plane SAW displacement. Images of SAWs incident on fabricated PnC at **a** 33 MHz and **b** 97 MHz. The waves are excited far left of the image propagating right, the dark strip in the centre is the area of fabricated PnC. At 33 MHz the SAW propagates through, at 97 MHz the PnC has a bandgap

higher frequency peaks correspond to their harmonics. These transducers are used, over chirped transducers for example, because they are commercially available and can be used for delay lines of large propagation distance. The transducers are arranged so that propagating SAWs are excited in the [100] direction of the PnC, corresponding to modes along the dashed black line in the simulated band diagram for wavevectors between the Γ and X in Fig. 1. In Fig. 4b the PnC-induced attenuation, relative to the unpatterned devices, is plotted as a function of frequency. These values were obtained by extracting the peak amplitude at each transducer resonance and calculating the attenuation as $20 \log(A_{patterned}/A_{blank})$ where $A$ is the detected SAW amplitude in mV.

The measured attenuation is approximately 3.3, 2.9 and 6.0 dB for SAWs excited at frequencies of 11, 33 and 120 MHz, and is much smaller at these frequencies than at any others. This is consistent with the simulations that show that at these frequencies propagating SAW bands are both present and available to couple to with transducer-excited SAWs in the [100] direction in Fig. 1, meaning SAWs at these frequency are expected to be able to propagate through the PnC. For infinite arrays, the coupling of modes is only possible when they are matched in frequency and in momentum. However, in this case, with a finite sized array, momentum is not well defined and, as such, coupling is possible without the bands crossing: the efficiency of coupling into the PnC modes is then dependent on the difference in momentum at a defined frequency. At 11, 33 and 120 MHz the Rayleigh dispersion is close in momentum to propagating PnC bands, but not matched. This is the reason the attenuation is small but non-zero.

The measured attenuation at a SAW frequency of 97 MHz, 12.6 dB, is much higher which again is consistent with the simulations as this frequency is expected to lie within a bandgap of the PnC, and therefore propagating SAWs should be attenuated by the crystal. Note that the discrepancy between the value of attenuation obtained from these measurements and that obtained from the simulations, 24.5 dB, is likely because the electrical value comes from measurements made on separate blank and PnC devices. As a consequence, the value of bandgap attenuation taken from the electrical measurements will be susceptible to differences in the electrical characteristics of the devices, caused by variations in bonding and mounting, etc. SAWs, excited at the other frequency harmonics of the transducer, lie outside the soundline within the first Brillouin zone. For this reason, SAWs at these frequencies are expected to be attenuated through the loss of energy to BAW coupling. This is again reflected in the measurements, where SAWs excited at frequencies above 120 MHz are all attenuated by at least 14 dB.

**Laser vibrometry measurements**. To verify these electrical measurements, the SAWs propagating in the PnC device were probed using the laser vibrometry method described by Smith et al.[33], which detects out of plane displacement through the measurement of the intensity and angle of a reflected laser spot incident on the substrate. Figure 5a, b shows the measured SAW displacement over the annular hole PnC at frequencies 33 and 97 MHz respectively, where SAWs have been excited continuously to the left of the image with the same transducer as in the electrical measurements described earlier. The colour scale represents a time-averaged amplitude of the SAWs. The dark strip in the centre of each image is the area of patterned annular hole PnC. Note that absolute values of the SAW displacement cannot be extracted from these measurements because the measured displacement is highly sensitive to device composition, in particular the reflectivity of the substrates. Even though the presented results are not absolute values, the measurements at different frequencies were performed on the same device under the same conditions, allowing the displacements to be compared.

As the measurements are time-averaged, the displacement of the propagating SAW results in a spatially uniform response from the system, whereas the presence of standing waves will result in a spatially modulated measured displacement amplitude. This can clearly be seen in Fig. 5a where there is a relatively small spatially modulated variation in displacement, along the x-direction, superimposed on a much larger constant background. As expected, at this frequency the propagating and standing wave SAWs are present both before and after the PnC, confirming that there is little attenuation of the SAW away from the bandgap frequency. In this case, the relatively small standing waves are a result of reflections from the device boundaries and transducers. There is also some small modulation in the y-direction which are artefacts of the raster scanning technique used.

The data obtained at the bandgap frequency, Fig. 5b is strikingly different where, as expected, there is almost no propagating SAW beyond the PnC, whereas there are relatively large standing waves to the left-hand-side of the crystal (where the propagating SAW is incident). The standing wave displacement amplitude before the PnC is 4.0 dB greater at 97 MHz than at 33 MHz, due to reflections of the SAWs from the crystal at the bandgap frequency. In addition, the observed reflection pattern is complex, due to the geometry of the crystal. Further investigations of these reflections are underway, but are beyond the scope of the work reported in this manuscript.

Values of the attenuation can be extracted from these measurements by taking the average displacement amplitude of the SAW before and after the PnC. The measured attenuation at 33 MHz is 2.67 dB, similar to the result found in the electrical measurements. At the bandgap frequency of 97 MHz, the value of attenuation extracted, 24.4 dB, is in excellent agreement with the value of 24.5 dB obtained from the simulations.

## Discussion

In conclusion, the novel concept of finite depth annular holes has been computationally and experimentally demonstrated to work as a SAW PnC, with low frequency bandgaps within the sound-line and up to an order-of-magnitude improved bandgap extinction compared to pillar PnCs. Simulations have shown the ability of the approach to tailor bandgaps by controlling the geometry to induce locally resonant modes. A square array of holes covering the aperture of a lithium niobate SAW delay line was fabricated using FIB etching, and the transmission of SAWs across the PnC was characterised as a function of SAW frequency. The minimum transmitted SAW amplitude was measured at a SAW frequency that lies in the bandgap of the PnC. These results

were verified with optical imaging of SAW displacement amplitude over the SAW delay line PnC device.

The annular hole PnC's high bandgap extinction for small numbers of features lends itself to many acoustic confinement applications, for which there is a broad range. In microfluidics SAWs are used for mixing, separating and other particle manipulations in drops and fluid channels, acoustic confinement could lead to the viability of multiple devices on a single chip[34–36]. This miniaturisation also applies to the biological application of an ultrasonically driven centrifuge[37] and similarly a SAW solid-state rotational micromotor[38]. Another field that could benefit from annular holes is that of thermal phononics in which applications include thermal transfer, thermal diodes, heat-cloaking[39] and pillar PnCs have been shown to achieve ultralow thermal conductivities[40]. The enhanced control of bandgap frequency and high in-plane symmetry also lend our annular hole approach for use in optomechanical applications such as phononic-photonic crystal cavities. These have been proposed as a pathway towards hybrid-quantum systems and integrations with 2D materials[41, 42].

## Methods

**Simulation methods.** Finite element method package COMSOL Multiphysics was used to find the PnC dispersions and relative transmission compared to pillar PnCs. The model simulating dispersions was developed from Assouar et al.[19] where a 3D unit cell with periodic boundary conditions according to the Bloch-Floquet theorem assumes an infinitely large square array. The bottom surface has a fixed constraint boundary condition to eliminate Lamb modes. An eigenvalue frequency domain solver was used with swept wavevectors within the first Brillouin zone to build the band diagrams. The model simulating relative transmissions was developed from Khelif et al.[21] where a 3D domain is finite in length in the propagation direction and has periodic boundary conditions in the perpendicular direction. Reflectionless boundaries reduce acoustic reflections, a piezoelectric transducer source is used to excite SAWs and a frequency domain solver is used for swept frequencies.

**Fabrication methods.** Annular holes were fabricated in lithium niobate. First a 400 nm layer of PMMA polymer was spin coated onto the substrate followed by a 100 nm thermal evaporated aluminium layer. The substrate was then focus ion beam etched with the metal layer grounded. The depth of etched holes was characterised by depositing platinum into them, etching a square trench across the hole and imaging the cross-section using oblique scanning electron microscope imaging.

**Experimental methods.** A SAW delay line of 128° YX-Lithium niobate with acoustic path length of 5.4 mm, two identical, double-electrode input/output transducers and aperture of 3.25 mm was used to characterise the PnC. Devices were mounted on a printed circuit board using a conductive silver epoxy and measurements were undertaken at room temperature, with the device mounted in a vacuum chamber, with measurements done at a pressure of ~$3 \times 10^{-6}$ mbar. Measurements were made by exciting a pulsed wave SAW at one transducer, using an Agilent 8648 C RF signal generator, and measuring the SAW amplitude at the opposing transducer using a LeCroyWaveRunner 204Xi-A digital oscilloscope. Results from a patterned device were compared to measurements made on several unpatterned, or blank devices. Acoustic imaging was done using a spatially resolved acoustic spectroscopy system employed by Renishaw detailed in ref. [33].

**Data availability.** The data that support the findings of this study are available from the corresponding author upon reasonable request.

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

## Acknowledgements

The authors acknowledge Renishaw plc and J. Coulson for providing use of the spatially resolved acoustic spectroscopy equipment and S.A.R. Horsley for helpful discussions. B.J.A. acknowledges funding from the EPSRC Centre for Doctoral Training in Metamaterials, grant number EP/L015331/1.

## Author contributions

G.R.N. conceived the study. B.J.A. carried out the simulations, fabrication and characterisation with the assistance of S.R.W. and G.R.N. All authors contributed to the data analysis and discussions. B.J.A. and G.R.N. wrote the manuscript, with input from all authors.

## Additional information

**Competing interests:** The authors declare no competing financial interests.

