## [Peer Review File · Nature Communications]

Reviewers' comments:

Reviewer #1 (Remarks to the Author):

Review NCOMMS-16-29840

The authors report on the design, fabrication and characterisation of a two-dimensional so-called annular phononic crystal. Such structures allow for the control the propagation of sound on the surface of a solid. Such structures are envisioned to be useful for various applications. The authors in particular focus on surface acoustic waves (SAWs), which are commonly employed in radio frequency electronics.

The main point of novelty of this study is the so-called annular design. In this approach the established implementations of phononic crystals composed either of free standing pillars or etched holes are combined: They consider in the first part of their manuscript a periodic arrangement of rings etched into the surface of a LiNbO₃ SAW device. Using finite element modelling, the authors show that their approach promises superior performance than established pillar based phononic crystals. They use advanced focused ion beam milling to fabricate test devices. They also show experimentally, that there exists a phononic bandgap, i.e. the region of acoustic frequencies in which the propagation is inhibited.

This reviewer considers the design of an annular phononic crystal novel in this scientific community working on acoustic waves. Despite the significant progress in this field, the proposed structure may be expected to be very attractive and adopted in future studies.

This reviewer considers the reported work a borderline case and requests clarification of several points. He/she in particular raises the following concerns:

(1) The idea of an annular design was already put forward more than 10 years in work on annular photonic crystals by Hamza Kurt and D. S. Citrin, *Optics Express* 13, 10316 (2005). In the photonic, optical domain the fabrication of such annular structures is not as straightforward as for sound. Therefore, to this reviewer's knowledge, only little experimental progress was reported since this original proposal.

One of the very few example in fact is a annular photonic crystal made on LiNbO₃ by Si and coworkers (*J. Vac. Sci. Technol. B* 29, 021205 (2011)) fabricated by exactly the same approach as in this manuscript.

Both works have to be referenced and its relation to the presented study addressed. So far this is not the case in the manuscript.

(2) The focus is mostly on simulations. Since the design includes several geometric parameters (radii, depth, lattice constant), a schematic would be helpful, Moreover, the figures are not clear. Fig. 1 (c), for instance, the bandage limits are plotted as a function of hole radius. What is this radius? is it the inner or outer radius? Fig. 2 compares the annular design to a pillar design. Here a schematic would be very helpful. Also a supplementary figure similar to Fig. 1 should be included for the pillar design. Fig. 3 and 4 could also be combined because Fig. 3 contains very little information.

(3) In Fig. 5 the authors compare the measured transmission of SAWs through their annular phononic crystal and compare it to the plain, unpatterned surface. It would be much more convincing if the performance was compared to the pillar design as this is the central point of their simulation work. Did the authors perform such an reference measurement? If so it would be nice to include this data. If not, it should be pointed out why this was not possible, e.g. due to the fabrication technique.

(4) In Fig 5, the authors seem to probe only at discrete frequencies. Why is this the case? In ref. 16, a broadband probe was apparently used. Also in this context: the SAW technique is not sufficiently explained. Too little information is given on the design of the test device, in particular on the transducers.

(5) The authors also mention potential applications beyond routing acoustic signals on a chip. Are there further applications possible, in addition to 2D materials (ref. 26 and 27)? An extended discussion and putting this approach in a broader context might make the manuscript more appealing for a broader audience.

Are there any particular advantages of the annular design in this context compared to a pillar or hole design, for instance for single quantum systems located in the center pillar?

In summary, this reviewer recommends a thorough, major revision of the manuscript. If the revision meets the concerns raised, this reviewer expects that the manuscript could be publishable in Nature Communications.

Reviewer #2 (Remarks to the Author):

In this manuscript entitled "Highly Attenuating Bandgap, Tunable and Robust Annular Hole Phononic Crystal for Surface Acoustic Waves", the authors introduce a novel geometry consisting of finite depth annular holes, which allow local resonances to be supported, that can open band gap for SAW. The main result claimed by the authors is that the array of annular holes improves the extinction of the transmission by an order of magnitude compared to a pillar phononic crystal.

- Tunability should allow controlling the bandgap dynamically, which is not the case in the present work. I do not believe that the word tunable is appropriate. The variation of the bandgap features by changing the geometrical parameter is static effect and commonly observed in phononic crystals whatever the inclusions.
- The numerical simulations of the transmission feature normalized values exceeding unity, which is not correct as there is no gain in the system. This occurs particularly in the phononic crystal with annular holes. This is an important issue as it affects strongly the estimation of the extinction ratio, which is one of the key results of the manuscript.
- In the experimental results, there are no details on the IDT configuration and the electrical responses are puzzling. The authors should account clearly for the results shown in figure 5 a.
- The resolution of the optical characterization presented in figure 6 is very poor. This makes it hard to be convinced. In addition, it seems that there is a plot missing.

Based on these considerations, I don't believe that the present manuscript reaches the criteria of Nature Communications.

Reviewer #3 (Remarks to the Author):

see attached review.

Referee's report on:

Highly Attenuating Bandgap, Tunable and Robust Annular Hole Phononic Crystal for Surface Acoustic Waves

by B.J. Ash et al.

The present paper describes a novel material structure that provides the ability to attenuate surface acoustic waves (SAWs) via a low frequency resonance. This is certainly a novel structure and to my knowledge this resonant structure has not been used before in this context. Based on this and the fact that the results are interesting and have been confirmed both experimentally and numerically it seems that the paper can be published. I just have a few comments below that the authors should take into account before publication:

My other points are the following:

- The title doesn't quite make sense - Either "A" is needed at the start or "Crystals".
- In the opening sentence "Phonon bandgaps" are mentioned. One must be careful in this context. Although acoustic materials are known as "phononic" band gaps I always think this is rather dubious ground, especially regarding the connectivity with heat and phonons.....perhaps some clarification is in order.
- L50. Although soundline is a relatively common term, it may be worth defining a little more clearly.
- The authors need to be careful with the word "tuning". Although this was used when phononic materials were first developed, in more recent times the word tuning has been used more proactively in terms of tuning phononic behaviour in real time by applying e.g. elastic, electric or magnetic fields, see e.g. the groups of Katia Bertoldi at Harvard and William Parnell and Manchester.
- There is a typo in Fig 1. - "wave vector"

We have addressed the specific points raised by the reviews below, and have high-lighted changes made to the text in the revised manuscript in yellow:

Referee 1

Comment 1.1: We thank the reviewer for highlighting the previous research on annular photonic crystals and have added references to the papers mentioned.

Comment 1.2: To clarify the geometric parameters associated with our design, we have added a schematic diagram of the unit cell in Figure 1(c), have relabelled the x-axis of this figure as “inner radius’ to remove any ambiguity, and have inserted a similar schematic in Figure 2. We have also added a figure similar to Figure 1, but for pillars, in the supplementary information. As suggested by the referee, we have also combined Figures 3 and 4.

Comment 1.3: Unfortunately, it has not been possible to do reference measurements on pillar phononic crystal for the reasons alluded to by the reviewer - the fabrication of LiNbO_3 pillars is extremely difficult and requires a different method than the focussed ion beam used here (see reference 20). However, the properties of pillar-based phononic crystal have been extensively explored over the last few years and we are confident that extra measurements would not change the conclusions of our work. As the relative ease of fabrication is a clear advantage of our design, we have added text to the experimental results section (lines 158-163) to highlight this advantage, and also state that reference measurements were not possible (line 175-176),

Comment 1.4: For this work we used commercially available SAW delay lines which contain double-electrode interdigital transducers. The transducers can excite a number of harmonics, giving rise to the discrete frequencies noted by the reviewer, and can be used to excite and detect SAWs more efficiently than a broadband transducer (e.g. one that is chirped). This greater transducer efficiency allows for the excitation of higher intensity SAWs which are easier to detect optically, leading to the results shown in Figure 5, and the fabrication of SAW devices with much longer propagation distances. We have added a more detailed description of the transducers, together with the rationale for using them, in the experimental results section (lines 176-183).

Comment 1.5: In addition to the integration with 2D materials also noted, a further potential application for our approach is to use it for optomechanical cavities associated with hybrid quantum systems. Applications in this area require a crystal with both phononic and photonic bandgaps and, as detailed in ref. 33, a structure with high in-plane symmetry, good bandgap frequency control and a planar surface: these properties and attributes are all possessed by the annular hole system described in this work. Text that details this in the discussion section of the manuscript has been expanded to elucidate more specifically the future ways in which the system can be exploited (239-242).

Referee 2

Comment 2.1: On reflection, we agree with the reviewer that the term “tunable” is potentially misleading and have replaced it with ‘tailorable’ or ‘frequency tailorable’ throughout the manuscript. The meaning intended is that, in order to tailor SAW dispersions, an annular hole system has a greater number of geometric degrees of freedom to change than a pillar-based system. The supporting text for this has been expanded in the introduction (lines 70-72).

Comment 2.2: The reviewer correctly highlights that the normalised transmission (Figure 2) cannot exceed unity. For both the case of the pillars and annular holes, the transmission is referenced with respect to a blank device that has no patterning so that, at frequencies away from the bandgap frequency, the normalised transmission should be unity (or zero on a log scale). In our original simulations, the normalised transmission exceeded unity as a result of spurious reflections of the acoustic wave arising from the way the model was discretised at the boundaries. As a result of the reviewer’s comment, we have re-run the simulations, this time adopting a Floquet discretisation approach, and we have plotted the results obtained in a new version of Figure 2. As can be seen, the transmission no longer exceeds unity at any frequency, whereas the difference in the attenuation at the bandgap is unchanged, with an order of magnitude smaller transmission in the annual hole array compared to the pillar array. We are grateful to the reviewer for flagging the issue and recommending we address the concern. We believe the improvement in our data confirms the attractiveness of our pillar-based architecture.

Comment 2.3: We have added more details regarding the IDT configuration (see **comment 1.4** above).

Comment 2.4: The spatial resolution of the acoustic spectroscopy system used to obtain the data presented in Figure 6 is estimated to be approximately 20 μm in the direction of the SAW propagation (see reference 32). The spatial resolution is therefore more than sufficient to compare the propagation of the SAW at frequencies lying outside and within the phononic bandgap (Figure 6a and 8b respectively). Unfortunately, we included a figure caption from an earlier version of the manuscript and apologise for the confusion caused, and have corrected the figure and caption.

Referee 3

Comment 3.1: We have amended the title of the manuscript to “A Highly Attenuating and Frequency Tailorable Annular Hole Phononic Crystal for Surface Acoustic Waves” (see also **comment 2.1** above).

Comment 3.2: Although we share the reservations of the referee, the term “phononic bandgap” has become standard in the literature for this type of structure. For consistency, we have replaced “phonon bandgap” with “phononic bandgap” (line 27).

Comment 3.3: We have included a fuller definition of “soundline” in the introduction (lines 48-51).

Comment 3.4: We agree with the reviewer that the term “tunable” is potentially misleading and have replaced it with ‘tailorable’ or ‘frequency tailorable’ throughout the manuscript (see comment 2.1 above).

Comment 3.5: We have corrected the typo on Figure 1.

Reviewers' comments:

Reviewer #1 (Remarks to the Author):

In the revised version the authors addressed most points raised by the three reviewers.

I can now recommend acceptance.

However I encourage the authors to expand the discussion section including further implications and fields of applications with corresponding references. (Comment 1.5 in their rebuttal letter). I assume that there are more than the two examples discussed so far.

This minor improvement does not require another round of peer review.

Reviewer #2 (Remarks to the Author):

This new version proposed by the authors and the responses made to the different reviewers' comments are unfortunately not convincing enough and do not allow this paper to reach the criteria for publication in Nature Communications. Even if the basic idea behind this paper, that is, to use an annular design for the inclusions of a phononic crystal is definitely interesting, the claims made by the authors are insufficiently supported both numerically and experimentally.

- The new title, that mentions tailorability and not tunability, is indeed more accurate and reflects more correctly the actual contribution of this manuscript to the field.

- The new simulation results seem more correct, as the normalized responses do not exceed unity any longer. Yet, the claim of the authors made in the abstract of "band gap attenuation one order of magnitude larger than that achieved with a pillar phononic crystal" is absolutely not convincing:
* the attenuation obtained in a pillar phononic crystal depends on the pillar height. A genuine comparison should take this into account by considering different geometrical parameters for the pillar phononic crystal;

* even in the case considered here, the attenuation is only significantly larger for a single-period wide phononic crystal, which does not make sense when considering an artificial periodic structures. The difference in attenuation drops to only a few dB as soon as the crystal width reaches two periods.

- The electrical characterizations still raise some questions. The higher-order harmonics of the surface wave device exhibits a higher response than the fundamental mode, which is definitely surprising, particularly for a commercial device. There is no comment made on the fact that the experimental attenuation is much lower than numerically predicted. There is a single frequency lying within the band gap, which makes a full analysis difficult.

- The response of the authors regarding the optical characterization section (point 2.4) is not convincing enough. At 110 MHz, the wavelength is about 38 μm , hence a resolution of 20 μm cannot be considered as "more than sufficient", as stated by the authors. The measurements in the phononic band gap case are hardly above the noise level. The quality of the measurements is indeed very poor. In Fig. 5b, one would expect a higher displacement amplitude due to the total reflection in the band gap. A standing wave pattern should appear as well. The use of arbitrary units prevents the reader from getting a fair idea of the measured signal.

Based on these considerations, I do not believe that this manuscript warrants publication in Nature Communications.

We have addressed the specific points raised by the reviewers below, and have high-lighted changes made to the text in the revised manuscript in yellow:

Referee 1

Comment 1.1: We have expanded the discussion section, lines 279-290, with additional applications of the phononic crystal, including; the use in microfluidics, thermal phononics, optomechanical systems, and micro-robotics.

Referee 2

Comment 2.1: The referee raises an interesting point regarding the dependence of the attenuation of a pillar phononic crystal on the pillar geometry. However, this dependence is not linear, as briefly demonstrated by Achaoui *et al*²⁵, and to our knowledge has not been fully explored previously. The majority of articles which explore the height of a pillar typically only present the bandgap frequency dependence, and not attenuation dependence. A thorough study, both theoretical and experimental, of the effect of pillar height and hole depth on bandgap attenuation would therefore be very interesting for future investigation, but is beyond the scope of this work.

However, to address the concern over whether our annular hole phononic crystal continues to outperform pillars for different geometrical parameters, we have run additional simulations for holes and pillars with different aspect ratios, both for 1 and 9 periods, as shown in Table 1.

Structure Depth/ Height	Attenuation (dB) Hole / Pillar		Frequency (MHz)
	1 period	9 periods	
5µm	21.4 / 8.9	38.3 / 38.0	120
6.4µm	20.0 / 8.4	26.9 / 22.8	95
9µm	14.4 / 8.1	18.8 / 20.8	57

Table 1 | Peak bandgap attenuations. Attenuation versus hole depth / pillar height for 1 and 9 periods of a phononic crystal with associated centre frequencies of bandgaps.

Our results show that the annular hole phononic crystal has a much higher bandgap attenuation than the pillar crystal in all but the most extreme, and impracticable, geometry. For 1 period, the increased attenuation of the annular hole vs the pillar is 12.5dB at 5µm and 6.3dB at 9µm, corresponding to an acoustic intensity ratio of 17.8 and 4.3 respectively. For 9 periods, the performance of the pillars only exceeds that of the annular holes for a height of 9µm.

However, the 9µm example is an extreme case with a height to radius aspect ratio of 2.2. The largest experimentally achieved aspect ratio for a pillar phononic crystal we have been able to find is 1.88, demonstrated by Yudistira *et al*²² using a domain inversion fabrication method only applicable for specific crystal cuts of a piezoelectric. The experimental case in our work has an aspect ratio of 1.56 which is comparable to many conventional pillar studies^{18-19, 21, 25}.

The results shown in Table 1 also highlights the referee’s observation that annular holes have a much higher peak attenuation relative to pillars for a small number of elements. This not only provides a route towards much more compact devices, but is also particularly important for the realisation of elements such as optomechanical cavities⁴¹, where the separation of the cavities is often critical. The annular hole phononic crystal described here could be used to create effective acoustic cavities that are only separated by a few crystal periods. Fig. 2b shows that for a phononic crystal with achievable geometric parameters, in a typical frequency range for surface acoustic wave devices, 1 period of an annular hole matches the bandgap attenuation of 7 periods of a pillar phononic crystal, as is already mentioned in the manuscript.

Coupled with the fact that smaller height to width aspect ratios are in general much easier to fabricate, these new simulation results further confirm the potential of the novel annular structure to transform the ability to exploit phononic crystals for developing novel SAW devices.

We have added Table 1 to the supplementary information, and have inserted a discussion on the effect of structure aspect ratio in the manuscript (lines 154-161). To caveat the fact that the order of magnitude improved

bandgap attenuation is not applicable universally, the term 'up to an order of magnitude' has been added to the abstract and discussion.

Comment 2.2: The higher response of higher order harmonics is indeed a feature of the commercially available devices used, the manufacturer's spec sheet can be provided on request.

An expanded discussion of experimental attenuation compared to simulated attenuation has also been added to the manuscript lines 212 – 220, see discussion below.

Comment 2.3: We want to emphasise that the main purpose of undertaking the laser vibrometry measurements, shown in Fig. 5, was to further demonstrate that the annular phononic crystal effectively prevents the propagation of the SAW at the bandgap frequency. As the measurements are time-averaged, the displacement of the propagating SAW results in a spatially uniform response from the system. In the original version of Fig. 5, we had used nearest neighbour averaging to allow us to better compare the displacement of the propagating SAW before and after the PnC.

We have now removed this averaging and created a new version of the figure which, as highlighted by the referee, allows standing waves to be seen (although it should be noted that these are superimposed on a spatially constant background due to the propagating waves). Note also that we have reversed the x-direction of the figure so that the SAWs are propagating from left to right, which we believe is somewhat more intuitive.

As predicted by the referee, at the bandgap frequency much larger standing waves are observed, but only to the left-hand-side of the PnC, indicating that these are primarily due to reflections from the crystal. The reflection pattern is complex, as might be expected from the geometry of the crystal. Further investigations of these reflections are underway, but are beyond the scope of the work reported in this manuscript. Note that although absolute values of the SAW displacement cannot be extracted from these measurements, as the measured displacement is highly sensitive to the specific device being measured and in particular the reflectivity of the substrates, the measurements at different frequencies were performed on the same device under the same conditions, allowing the displacements to be compared.

We have therefore now extracted a value of the bandgap attenuation, using the average measured displacement before and after the PnC, of 24.4 dB, which is in very good agreement with the value of 24.5 dB obtained from the simulations. Note that the discrepancy between these values and that obtained from the electrical measurements, 12.6 dB, is likely because the electrical value comes from measurements made on separate blank and phononic crystal devices. As a consequence, the value of bandgap attenuation taken from the electrical measurements will be susceptible to differences in the electrical characteristics of the devices caused by variations in bonding etc. We have added a few sentence to this effect from line 212. This is in contrast to the optical measurements, which were made on the same device.

The experimental section from line 226 has been revised with these additional results and explanations as discussed above, and we thank the reviewer for helping us better interpret this experimental data.

REVIEWERS' COMMENTS:

Reviewer #2 (Remarks to the Author):

The last version of the manuscript contains the whole information to explain the results. The authors have answer to the asked questions. It is a nice piece of work and the results is interesting for the phononic community. However, I still believe that there no new mechanism or a technological performance that allow a publication in Nature communication. In this form, the work is more suitable for publication in Scientific reports.

Response to Referees

We wish to thank the reviewer for taking the time to review our manuscript for a third time, and for acknowledging that we have now addressed all the technical issues raised. We have therefore not made any changes to our manuscript in light of the reviewer's comments. As stated previously, our belief in the importance and timeliness of this work, and its suitability for publication in Nature Communications, has been considerably strengthened by the overall review process.